# Comment on Costa et al. The Effectiveness of Different Concepts of Bracing in Adolescent Idiopathic Scoliosis (AIS): A Systematic Review and Meta-Analysis. *J. Clin. Med.* 2021, *10*, 2145

**DOI:** 10.3390/jcm11030752

**Published:** 2022-01-30

**Authors:** Hans-Rudolf Weiss

**Affiliations:** Schroth Best Practice Academy, 55546 Neu-Bamberg, Germany; hr.weiss@koob-skoliose.com

I read the above-mentioned work with great interest, and I would like to thank the authors for considering two papers from our working group. However, I wonder why our paper containing the preliminary results of the treatment with the Gensingen Brace [1] was taken into account within this systematic review but not the final results that were first published online in 2020 [2]. The latter paper with the final results would have had a lower risk of bias than the paper containing the preliminary results.

The authors seem committed to finding the most effective concept of bracing or even the most effective brace model. In my opinion, however, they have chosen the wrong study design for this because one cannot find this through averaging and can only find it through an individual case analysis.

Surprisingly, in their systematic review regarding night-time brace treatment, the authors came to completely different conclusions than the authors of another current systematic review on the subject [3]. Ruffilli et al. [3] concluded that “*the current available literature does not permit us to draw conclusions about night-time braces. The low methodological quality of the studies examined makes it impossible to compare the effectiveness of the night-time braces with that of traditional TLSOs*”.

In fact, the results of night-time bracing are very diverse. As early as 1997, there was a meta-analysis that showed that part-time or night bracing was clearly inferior to full-time brace treatment [4]. Further studies on this topic clearly show that wearing time and the correction effect are the two main criteria for success [5,6]. The greater the corrective effect of a brace and the longer the wearing time, the greater the success rate [5,6]. However, the authors do not discuss these important parameters in their study.

The question is how the authors of this systematic review, compared to Ruffilli et al. [3], came to such different conclusions:

One study on the Providence Brace included by the authors has an exceptionally high success rate [7], while other studies on the Providence Brace tend to conclude with success rates of between 50% and 60% [8]. One study cited by the authors has a success rate of over 70%, but this is a more mature and therefore prognostically more favorable cohort (Risser 0–3) [9]. 

A success rate of 89% in a night-time brace is very exceptional [7]. However, if one reads the study carefully, one will find the following passage on the subject of inclusion criteria: “*We included all patients, diagnosed with AIS who fulfilled the following criteria, age >10 years at time of diagnosis, less than 12 months post-menarche, Cobb 20°–42°, no prior scoliosis treatment, initial in-brace curve correction >60% and follow-up including radiographs at least 12 months after brace termination. The patients were braced according to the SRS criteria*”.

Accordingly, this study [7] is a selective study with favorable inclusion criteria. No statement can be found in the paper on the outcome of the patients with a correction effect of <60°. Interestingly, this particular patient selection is addressed neither in the abstract nor further elaborated in the discussion. In addition, the collective of Simony et al. [7] also failed to meet the SRS inclusion criteria for brace studies. The Risser sign, as an essential part of the SRS inclusion criteria, was not recorded, and the cohort was also significantly older than in comparable studies on the topic [1,2]. This was—among other things—already discussed in a letter to the editor by Dr. Potts [10].

The problem was thus known, and yet, this work with a considerable selection bias [7] was included by the authors in their systematic review with meta-analysis and was related to other studies without correspondingly favorable patient selection. This systematic review with meta-analysis by Costa et al., therefore, has a systematic selection bias and should not have been published in the present form. This all too positive outlier with a success rate of 89% [7] has a significant influence on the results of this review. Without the inclusion of the Simony paper [7], one would not come to the conclusion that the results of night-time bracing are on a par with full-time bracing.

Finally, the authors did not indicate a conflict of interest (René M. Castelein, Stryker Spine (a, d), found in the programs of the SRS Conferences 2020 and 2021, [11,12]).

The fact that publications contaminated by errors coexist with their healthy counterparts in different databases, and in the worst-case scenario, multiply in systematic reviews and meta-analyses, is well recognized [13]. And this is just what has happened in the paper addressed within this comment.

## Conclusions

The goal of finding the most effective bracing concept or even the most effective brace model was not achieved and was not further discussed in this work.The unveiled findings cannot be reconciled with the study design of a systematic review with meta-analysis. The study shows features of a narrative review with a pronounced selection bias, which significantly influences the conclusions.The fact that the described original work with a considerable selection bias [7] was included by the authors in their so-called systematic review with meta-analysis despite extensive discussion in the literature [9] raises doubts about a simple oversight as the cause.

## Data Availability

Not applicable.

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
