# Peer review of "Comment on Costa et al. The Effectiveness of Different Concepts of Bracing in Adolescent Idiopathic Scoliosis (AIS): A Systematic Review and Meta-Analysis. J. Clin. Med. 2021, 10, 2145"

_jcm, 2022, doi:10.3390/jcm11030752_

Round 1
Reviewer 1 Report
The Author reports and justify with clarity his point of view about the mentioned systematic review. His thesis is well supported by the referenced manuscript and provide a necessary clarification about a controversial matter.
Author Response
Thanks a lot for your appraisal. Although the ms. has been overworked by a native English speaker I have made another spell check.
Reviewer 2 Report
This letter shows important drowbacks of the published paper. In addition shows the conflict of interest that was not disclosed.
The paper should be published. Some parts of the letter could be more concist.
The fragment 'However, I wonder why our work with the end results of the treatment with the Gensingen Brace was not also taken into account in this systematic review [1]. The online version of this work was already available in Pub Med before the end date of their review carried out by the authors, namely in October 2020. In addition, this work with the final results would have had a lower risk of bias than our work with the preliminary results [2]'. Please concidere modification. Just state that analysis inclused preliminary, but not final data from your group.
The statement 'The question now is how the authors of this systematic review, compared to Ruffilli et al. [3] came to such different conclusions.' should be modified (less agressive). For instance: Suprisingly the authors came to different colcusions based of quite similar studies. Plese describe shortly the differences in the included studies. Please stress the bias.
Please shorten the fraze: 'Finally, it should be noted that the authors did not indicate a conflict of interest although at least the latter author has a connection to industry (René M. Castelein, MD, PhD, Stryker Spine (a, d); to be found in the program of this year's SRS Conference / https://www.srs.org/UserFiles/file/AM21_FinalProgram.pdf).'
Perhaps sugest the bias related to undisclosed conflict of interes of the study authors related to the colaboration with industry. Do you have convincing prove for the collaboration prior to publication?
Author Response
I wish to thank the reviewer for the chance to improve my letter. In the following you will find a point to point response:
- the § has been changed accordingly and the refs 1 and 2 have been exchanged:
However, I wonder why our paper containing the preliminary results of the treatment with the Gensingen Brace was taken into account within this systematic review [1] but not the final results as published online first in 2020 [2]. The latter paper with the final results would have had a lower risk of bias than the paper containing the preliminary results.
2. § changed:
Surprisingly the authors in their systematic review regarding the night-time brace treatment came to completely different conclusions than the authors of another current systematic review on the subject [3]. Ruffilli et al. [3] concluded that ‘the current available literature does not permit us to draw conclusions about night-time braces. The low methodological quality of the studies examined makes it impossible to compare the effectiveness of the night-time braces with that of traditional TLSOs.’
3. Ref added:
One study on the Providence Brace included by the authors has an exceptional high success rate [7] while other studies on the Providence Brace tend to conclude with success rates of between 50 and 60% [8]. One... 4. Phrase shortened and the links were removed from the text to the ref section. Besides the 2021 program the same was found in the 2020 program as evidence that there was this CoI before publication of the paper:Finally, the authors did not indicate a conflict of interest (René M. Castelein, MD, PhD, Stryker Spine (a, d); to be found in the programs of the SRS Conferences 2020 and 2021; [11,12]).
5. A short § was added with a citation expressing exactly the issues as described within this letter. Interestingly, within this citation a table can be found with reasons for retraction. More than one reason for retraction of the paper in question can be found:
The fact that ‚publications contaminated by errors coexist with their healthy counterparts in different databases, and in the worst case scenario, multiply in systematic reviews and meta-analyses’ is well recognized [13]. And this is just what has happened in the paper addressed within this letter.

Round 2
Reviewer 2 Report
Not it is less agressive and more balanced.